# Self-Supervised Learning of Object Motion Through Adversarial Video Prediction

## Abstract

Can we build models that automatically learn about object motion from raw, unlabeled videos? In this paper, we study the problem of multi-step video prediction, where the goal is to predict a sequence of future frames conditioned on a short context. We focus specifically on two aspects of video prediction: accurately modeling object motion, and producing naturalistic image predictions. Our model is based on a flow-based generator network with a discriminator used to improve prediction quality. The implicit flow in the generator can be examined to determine its accuracy, and the predicted images can be evaluated for image quality. We argue that these two metrics are critical for understanding whether the model has effectively learned object motion, and propose a novel evaluation benchmark based on ground truth object flow. Our network achieves state-of-the-art results in terms of both the realism of the predicted images, as determined by human judges, and the accuracy of the predicted flow. Videos and full results can be viewed on the supplementary website: https://sites.google.com/site/omvideoprediction.

## 1 Introduction

When we interact with objects in our environment, we can easily imagine the consequences of our actions: if we push on a ball, it will roll and ricochet off of obstacles; if we drop a vase, it will break. The ability to imagine future outcomes provides an appealing avenue for learning about the world: unlabeled video sequences can be gathered autonomously with no human intervention (see Figure 1), and a machine that learns to accurately predict future events and the outcomes of its actions will have gained an in-depth and functional understanding of its physical environment. This leads naturally to the problem of video prediction: given a sequence of context frames, and optionally a proposed action sequence, generate the raw pixels of the future frames to predict how the world will evolve. Once trained, such a model could be used to determine which actions can bring about desired outcomes (Finn et al., 2016;

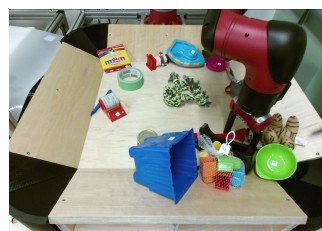

Figure 1: The video prediciton model is trained on 17,000 video sequences collected autonomously by a Sawyer robot interacting with 140 different every-day objects.

Ebert et al., 2017). Unfortunately, accurate and naturalistic video prediction remains an open problem, with state-of-the-art methods producing high-quality predictions only one or a few frames into the future.

Synthesizing videos offers a multitude of challenges. The natural image "manifold" is extremely thin relative to the space of pixels, and image generation is an active and unsolved field of research (Goodfellow et al., 2014; Kingma & Welling, 2014; van den Oord et al., 2016). Furthermore, the evaluation of the quality of synthesized images is notoriously difficult (Ramanarayanan et al., 2007), with automated metrics such as PSNR and SSIM (Wang et al., 2004) not corresponding well to human perception (Ponomarenko et al., 2015).

In this paper, we propose a novel video prediction model that is specifically focused on accurately understanding object motion. We argue that motion, rather than appearance, is the critical component required to understand physical interaction. While prior work has examined flow-based models that implicitly model the motion of pixels (Finn et al., 2016), such methods typically produce highly

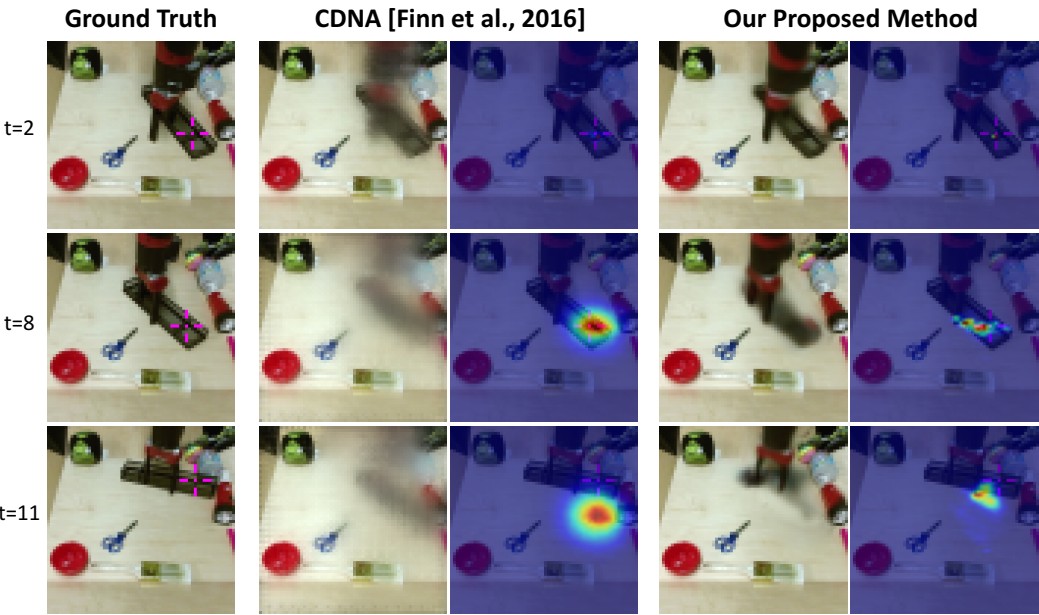

Figure 2: **Example results.** Our proposed system is capable of predicting complex long-term robot-object interactions. Left column: ground truth video with object position marked in the magenta cross. The heatmaps show the predicted probability distribution of the marked point for the CDNA model (Finn et al., 2016) and Our proposed method. Both accuracy (distribution mean) and precision (distribution variance) of our model are substantially higher than the baseline.

blurred predictions far into the future. We introduce a number of novel extensions to flow-based prediction that significantly improve prediction accuracy, both in terms of the appearance of the resulting images, and in terms of the quantitative accuracy of the predicted motion.

In the video prediction context, the uncertainty that the temporal dimension presents an additional complication. While immediate frames can typically be extrapolated with high precision (Mathieu et al., 2016), the space of possibilities diverges beyond a few frames, and the problem becomes multimodal by nature. Methods which use loss functions unequipped to handle this inherent uncertainty, such as Euclidean loss, will make average, blurry predictions. To this end, we explore using a *learnable* loss function, the GAN framework proposed by (Goodfellow et al., 2014) and made popular in the conditional setting in (Isola et al., 2017), which produces more realistic images. To our knowledge, our method is the first to demonstrate the use of an adversarial loss for multi-frame flow-based video prediction. Coupled with a novel flow-based architecture, this allows our model to produce substantially more naturalistic predictions up to 7 seconds into the future (corresponding to 15 frames).

Besides our improved flow-based prediction model, we also propose a novel metric for evaluating the quality of the predicted object motion. To decouple appearance from an understanding of physics, we propose a metric based on ground truth object motions, and discuss how this metric can be evaluated for models that implicitly predict flow as an intermediate step in video prediction.

Our contributions are as follows:

- We show that a learned discriminator can be paired with a flow-based generator, producing use a learned discriminator and show improvements in visual quality while slightly improving flow accuracy.

- We show large improvements in both visual realism and flow accuracy to the previous state-of-the-art methods in video prediction for robotic manipulation (Finn et al., 2016; Ebert et al., 2017).

- We present improved benchmarks for both flow accuracy prediction and visual realism. We propose a flow evaluation benchmark, containing ground truthed flows on real objects. We evaluate realism using a *real vs. fake* test using real human judges.

## 2    RELATED WORK

Approaches to video prediction and generation vary along two axes: the model for outputting pixels, and the objective used for training. Early approaches to video prediction focused primarily on models that generate pixels directly from the state of the model using both feedforward (Ranzato et al., 2014; Mathieu et al., 2016) and recurrent (Oh et al., 2015; Xingjian et al., 2015) architectures. In this work, we focus on transformation-based models of video prediction, such as those proposed by Finn et al. (2016); De Brabandere et al. (2016); Xue et al. (2016); Byravan & Fox (2016); van Amersfoort et al. (2017); Liu et al. (2017); Chen et al. (2017), which predict transformations from the current frame to the next. These transformations are then combined with the current frame prediction to produce the pixels of the future frame. A benefit of this approach is that it enables modeling of object motion, which can be useful in the context of control (Finn & Levine, 2017; Ebert et al., 2017). A similar approach is to predict optical flow directly (Walker et al., 2015; 2016), though this requires an accurate external optical flow solver, rather than only the raw videos. Our method does not require any supervision or labeling of the training data. Furthermore, in contrast to prior flow-based models, our approach incorporates an adversarial loss, which substantially improves the quality of multi-step predictions up to 7 seconds into the future.

Largely orthogonal to the choice of model is the choice of objective. Many methods have used the simple mean-squared error objective, which can lead to sharp predictions in deterministic synthetic settings (Oh et al., 2015; Chiappa et al., 2017), but cause the model to represent uncertainty as the average of the possible futures leading to blurry generations (Mathieu et al., 2016). There are three main approaches to altering the prediction objective to better model the distribution over future frames. One approach is to model the full joint distribution using pixel-autoregressive models (van den Oord et al., 2016; Kalchbrenner et al., 2017). While this can produce sharp images, training and inference are impractically slow, even when introducing parallelism (Reed et al., 2017), and these method do not explicitly model the object motion. Another approach is to use variational objectives, modeling the distribution over video frames using latent variables (Kingma & Welling, 2014; Watter et al., 2015; Xue et al., 2016; Johnson et al., 2016). While variational objectives can theoretically enable better modeling over the joint distribution over images, these methods have not been successfully applied to multi-frame prediction of real RGB images. The final approach is to use adversarial objectives (Goodfellow et al., 2014; Mathieu et al., 2016; Vondrick et al., 2016; Tulyakov et al., 2017). Similar to these papers, we use an adversarial objective, but we combine it with a transformation-based model which allows for direct modeling of object motion. To our knowledge, ours is the first method that effectively combines these two components in a practical multi-step video prediction framework.

One key challenge in video prediction is the lack of good metrics for evaluating predictions. Mathieu et al. (2016) evaluated on a range of objectives including PSNR, SSIM (Wang et al., 2004), and sharpness. While these metrics are easy to automatically measure, they do not provide good measures of image quality and usefulness (Ponomarenko et al., 2015). Others have used video prediction for representation learning, evaluating the learned representation for downstream tasks such as digit classification and pose prediction (Srivastava et al., 2015; Lotter et al., 2017; Denton & Birodkar, 2017). More recent approaches have used video prediction for planning motions on a real robot (Finn & Levine, 2017; Ebert et al., 2017; Byravan et al., 2017), which is a useful metric, but challenging to automate. In this work, we propose two procedures for evaluating video prediction models that measure the perceptive quality and object motion accuracy, respectively. These metrics are generic, in that they can be applied to any video prediction dataset, including real world videos, and require only a modest amount of effort to evaluate compared, for example, to executing object manipulation tasks on a real robot (Finn & Levine, 2017; Ebert et al., 2017).

## 3    VIDEO PREDICTION WITH FLOW AND ADVERSARIAL LOSSES

We describe our adversarial loss function and network architecture in this section. We first present the our overall network architecture, used for synthesizing future video frames by predicting flows. We then present the adversarial loss function. Finally, we describe additional architectural design decisions that enable our model to make accurate predictions of both object motion and object appearance.

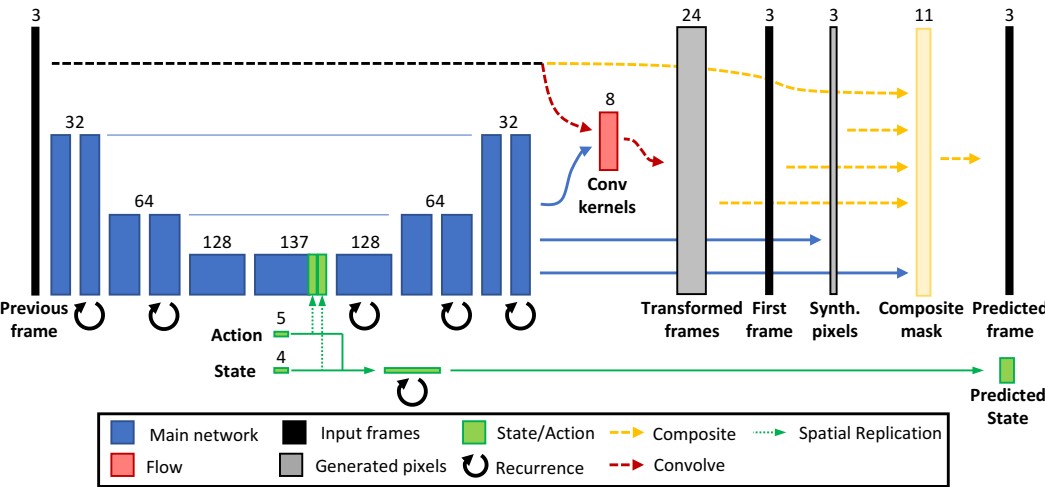

Figure 3: **Our network architecture**. Our network uses a U-Net (Ronneberger et al., 2015) with recurrent connections in internal layers (Schmidhuber, 1987). As proposed by Finn et al. (2016), the network predicts (1) a set of convolution kernels to produce a set of transformed input images (2) synthesized pixels at the input resolution and (3) a compositing mask. Using the mask, the network can choose how to composite together the set of warped pixels, the first frame, previous frame, and synthesized pixels.

**Architecture and recurrent prediction.** Our network architecture, shown in Figure 3, is inspired by the convolutional dynamic neural advection (CDNA) model proposed by Finn et al. (2016). The video prediction setting is a sequential prediction problem, so we we use a convolutional LSTM (Schmidhuber, 1987; Xingjian et al., 2015) to predict future frames. In our experiments, we initialize the prediction on the first two ground truth frames, and predict fifteen future frames, corresponding to 7.5 seconds into the future.

The model predicts a sequence of future frames by repeatedly making next-frame predictions and feeding those predictions back to itself. For each one-step prediction, the network has the freedom to choose to copy pixels from the previous frame, used transformed versions of the previous frame, or to synthesize pixels from scratch. The transformed versions of the frame are produced by convolving in the input image with predicted convolutional kernels, allowing for different shifted versions of the input. In more recent work, the first frame of the sequence is also given to the compositing layer (Ebert et al., 2017).

The motions are learned in a completely self-supervised way, with data acquired from the robot au-tomomously interacting with objects. Neither ground-truth flows nor labels are required for training. Note that the network predicts one-step flow distributions, which we denote as $\widehat{P}_{t\to t+1}$, which are an *emergent output* of our system. The predicted flows are per-pixel probability distributions, obtained from the transform convolutional kernels (which are probability distributions over pixel locations) and compositing process (which is probability distribution over different transformed frames). These can be iteratively composed together through time to predict the positions of objects at a future time step $\widehat{P}_t$.

**Adversarial loss function.** We propose using a GAN (Goodfellow et al., 2014) framework, with the transformation-based model in the previous section acting as the generator. To our knowledge, our work is the first to propose this formulation in the context of multi-frame conditional video prediction. For notational simplicity, we first describe the system in terms of 1-step prediction. Our generator, described in Equation 1, takes in the current image frame $I_t$, current state $s_t$ (in our application the end-effector pose of the robot), and current action $a_t$ and predicts the next frame $\widehat{I}_{t+1}$, state $\widehat{s}_{t+1}$, and flow transformation $\widehat{P}_{t\to t+1}$:

$$(\widehat{I}_{t+1}, \widehat{s}_{t+1}, \widehat{P}_{t\to t+1}) = \mathcal{G}(I_t, s_t, a_t) \tag{1}$$

The generator also implicitly ouputs a set of kernels $\widehat{T}_{t\to t+1}$ which the model internally uses to transform the previous image into the next image, and which can be leveraged to predict the position

of particular point over time. The conditional GAN framework uses a learned generator and discriminator in a minimax game. The generator tries to produce an image which can fool a discriminator, which is tasked with identifying fakes. From the point of view of the generator, the discriminator acts as a *learned* loss function. With perfect training dynamics, this forces the generator match all statistics of the real data distribution. The objective is shown in Equation 2, where $\mathcal{G}_I$ represents just the image output of the generator:

$$\mathcal{G}^* = \arg \min_{\mathcal{G}} \max_{\mathcal{D}} \ \log \mathcal{D}(I_{t+1}, I_t) + \log(1 - \mathcal{D}(\mathcal{G}_I(I_t, s_t, a_t), I_t)) \tag{2}$$

Training GANs can be difficult and is an active area of research. We adopt the `pix2pix` framework from Isola et al. (2017), which uses a per-pixel $\ell_1$ loss to stabilize training, and also use the LS-GAN (Mao et al., 2016) variation.

**Architecture improvements.** We made a variety of architectural improvements to the original CDNA (Finn et al., 2016), which produced better results in both the per-pixel loss and the adversarial loss cases. Each convolutional layer is followed by instance normalization Ulyanov et al. (2016) and ReLU activations. Empirically, instance normalization provided faster convergence for our model than layer normalization. The feed-forward convolution layers use $3 \times 3$ kernels, except for the first, which is $5 \times 5$, and LSTM layers use $5 \times 5$ kernels, with instance normalization on the pre-activations (i.e. the input, forget, and output gates, as well as the transformed and next cell of the LSTM). Note that this is different from the method proposed by Ba et al. (2016), which uses layer normalization in the outputs of the LSTM layers.

In addition, we modify the spatial downsampling and upsampling mechanisms. Standard subsampling and upsampling between convolutions is know to produce artifacts for dense image generation tasks (Odena et al., 2016; Zhao et al., 2017; Niklaus et al., 2017). In the encoding layers, we reduce the spatial resolution of the feature maps by average pooling, and in the decoding layers, we increase the resolution by using bilinear upsampling followed by convolutions.

**Training dataset.** We collected a dataset of 17792 training trajectories and 128 test trajectories. We withheld 896 trajectories from the training set for validation. Each trajectory consists of a sequence of images, states, and actions. The images are RGB images at $64 \times 64$ resolution, the states are 4-dimensional vectors consisting of cartesian 3-D positions of the gripper and the rotation angle of the gripper (around the gravity axis), and the actions are 5-dimensional vectors consisting of discrete-time velocities of the state and an additional indicating if the gripper opens or closes.

**Implementation details.** The network was initialized with truncated normals ($\sigma = 0.02$) for the non-bias weights and with zero biases. We trained it with Adam Kingma & Ba (2015) for 200000 iterations, with $\beta_1 = 0.9$, $\beta_2 = 0.999$, learning rate of $0.001$, and a batch size of 16. We observes that smaller batch sizes degrades the final convergence whereas larger batch sizes provides faster convergence in terms of iterations at the expense of overall training time. We used scheduled sampling during training as in Finn et al. (2016) so that at the beginning the model is trained for one-step predictions and by the end of the training the model is fully autoregressive.

## 4 EVALUATION METRICS AND EXPERIMENTS

Our experimental evaluation is aimed at comparatively studying the performance of our method versus prior models along two axes: the accuracy of the predicted object motion, and the realism of the generated future images. For our evaluation, we used a dataset of 17,000 video sequences collected autonomously by a Sawyer robot pushing 140 different objects in a tabletop setting. This dataset includes a wide variety of physical interactions, while at the same time providing for a controlled setting in which to evaluate the methods. We compare our method to the following variants and prior techniques:

**Ours.** Our full system, trained with both a per-pixel $\ell_1$ loss and an adversarial loss, with the network shown in Figure 3.

**Ours (no adversary).** Our network architecture trained only with a per-pixel $\ell_2$ regression loss, without the adversarial objective.

| Method | | Image Assesment | | | | Motion Accuracy |
|---|---|---|---|---|---|---|
| Model | Loss | PSNR (dB) | MSE ($\times 10^{-3}$) | SSIM | AMT Fooling (%) | Distance (pixels) |
| Ground truth | - | $\infty$ | 0.0 | 1.0 | 50.00 | 0.0 |
| CDNA (Finn et al., 2016) | $\ell_2$ | 22.7 | 6.72 | 0.828 | $9.48 \pm 1.04$ | $5.17 \pm 0.36$ |
| SNA (Ebert et al., 2017) | $\ell_2$ | 23.6 | 5.38 | 0.851 | $7.54 \pm 0.94$ | $4.11 \pm 0.32$ |
| Ours (no adversary) | $\ell_2$ | **26.7** | **2.70** | **0.916** | $16.87 \pm 1.40$ | **2.56$\pm$0.28** |
| Ours | $\ell_1 + \mathcal{D}$ | **26.6** | 2.91 | **0.916** | $\mathbf{25.19 \pm 1.59}$ | **2.45$\pm$0.29** |

Table 1: **Quantitative Results.** We show Image Assessment and Motion Accuracy metrics between our method and prior methods. We achieve higher scores on automated metrics (PSNR, MSE, SSIM) with respect to baseline methods, and similar scores with and without the adversary. When evaluated by real human judges (AMT Fooling), we see large improvements in realism. For motion accuracy, we see large improvements in our system relative to baseline methods, largely due to the architectural changes. Slight improvement in motion is observed by training with an adversary as well.

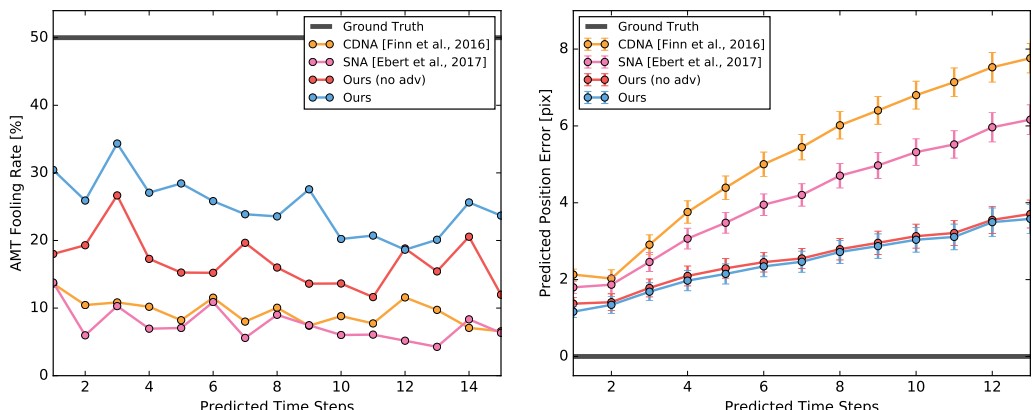

Figure 4: **Visual realism** (left) and **Predicted flow accuracy error** (right) through time. For visual realism (higher is better), we show that our architecture, trained with the same $\ell_2$ loss (**Ours (no adv)**) yields more realistic results than previous state-of-the-art (**CDNA, SNA**). Adding an adversarial loss (**Ours(adv)**) greatly improves visual realism. For flow accuracy error (lower is better), our architecture produces much more accurate flows than baselines, and adding the adversarial loss slightly improves accuracy.

**SNA** (Ebert et al., 2017). Prior flow-based model that uses an $\ell_2$ loss and temporal skip connections.

**CDNA** (Finn et al., 2016). Prior flow-based model that uses an $\ell_2$ loss.

## 4.1 EVALUATION METRICS

Evaluating the performance of a video prediction model is notoriously difficult. Automated metrics often do not provide an accurate measurement of the quality of the predicted frames. Perhaps more importantly, the basic question of what "quality" actually means is far from settled, and depends intimately on the intended eventual use of the prediction model. In this paper, we argue that two important criteria are the accuracy of the predicted object motion and the realism of the predicted images. Note that naturalistic images might not necessarily be close to the true images in terms of mean squared error – the motions of physical objects typically cannot be inferred with perfect accuracy just from raw images, which means that even the best predictive model can at best guess about one of a number of possible futures. Motivated by these criteria, we evaluate our model in comparison with prior methods along two dimensions: accuracy of the prediction motion and real human judgements of image realism.

**Motion accuracy.** Accurate motion prediction is a key requirement for a video-prediction to be useful for interacting with the physical world, for instance in the setting of robotic control. For ex-

ample, prior work (Ebert et al., 2017; Finn & Levine, 2017) proposed to combine video prediction models with a planning algorithm to enable a robot to manipulate physical objects in its environment. The performance of such real-world applications depends critically on the accuracy of the predicted motions. However, evaluating every model directly in the real world and on a real robot is time-consuming. Therefore, an accurate and reliable motion accuracy benchmark would greatly accelerate research progress on effective and useful video prediction. To that end, we propose a metric based on predicted object motion accuracy. To evaluate the accuracy of the flow predictions, we annotated object motion in 128 test videos. In each video, we manually labeled the positions of two points on two objects over the course of the sequence. We then compute the expected distance between the model's predicted object positions and the true positions as follows. Let $P_{t,d} \in \Delta^{H \times W}$ be the predicted probability distribution of a point $d$ on an object at a future time instant $t$, where $H = W = 64$ are our image dimensions. The expected distance to the true position $d^*$ is given by:

$$\delta_t = \mathbb{E}_{d_t \sim P_{t,d}} \|d_t - d^*\|_2 \tag{3}$$

All distances are in units of pixels (in images of 64x64). This expected distance provides a measure of the accuracy of the predicted object motion.

**Realism.** To evaluate the realism of the synthesized flows, we use real human judgments in a real vs. fake test, commonly used for evaluating synthesized images (Zhang et al., 2016; Isola et al., 2017; Zhu et al., 2017; Chen & Koltun, 2017). The test presents real and generated images to Amazon Mechanical Turker (AMT) workers for 1 second each, in a randomized order. The humans are then tasked with identifying the fake. An algorithm which produces completely unrealistic images would never fool the humans, and one which produces perfectly realistic images would achieve a 50% *fooling rate*. We use the publicly available implementation from Zhang et al. (2016).

## 4.2 EXPERIMENTAL RESULTS

The quantitative results of our experiments are summarized in Figure 4 and in Table 1. Furthermore, we include video results on the supplementary website: `sites.google.com/site/omvideoprediction`

**Does our algorithm synthesize accurate flows?** Figure 4 shows the average distance and standard error of the distance between the predicted and true object positions over time. As one would expect, for all methods the mean and uncertainty grows for predictions further into the future. The frame rate in the videos was 2 Hz, so the predictions are made out to 7.5 seconds into the future. The prediction error of our proposed model is significantly lower for all time steps, as compared to prior techniques. Table 1 shows the prediction errors averaged over the full trajectories. Our results indicate that the proposed video prediction model significantly outperforms prior work in terms of object motion accuracy. While the advantage of applying the adversarial loss (blue curve vs. red curve in Figure 4) is small, we hypothesize that the adversarial loss mainly affects the *appearance* of the predicted objects, having less impact on the estimated locations.

**Does our algorithm synthesize realistic images?** To acclimate the human evaluators to the robotic manipulation image domain, we provide 20 "training" test runs to each evaluator, in which they get immediate feedback on whether they successfully selected the fake image. The evaluators are then tested on 50 image pairs. A total of 234 trials were run for a total of 11,700 evaluations across the 4 algorithms.

As seen in Table 1, our method, trained with an $\ell_2$ loss, achieves a significantly higher fooling rate than previous work from that also uses an $\ell_2$ loss (Finn et al., 2016; Ebert et al., 2017). This indicates that the design of our architecture produced results that were more realistic, as determined by human observers. Furthermore, adding a learned discriminator loss increased the fooling rate from 16.9% to 25.2%. Figure 4 (right) shows the fooling rate as a function of how far a synthesized frame is in the future. Across all methods, the fooling rate decreases with respect to time, since future predictions gradually decrease in quality. The fooling rate of our system, even at the final time step, is higher than for all previous previous work. In fact, even comparing the fooling rate of our method at the last frame, we see that it is substantially higher than for the prior methods at the *first* frame, indicating that our architecture and adversarial loss are able both to obtain a high degree of realism and maintain it throughout the video sequence.

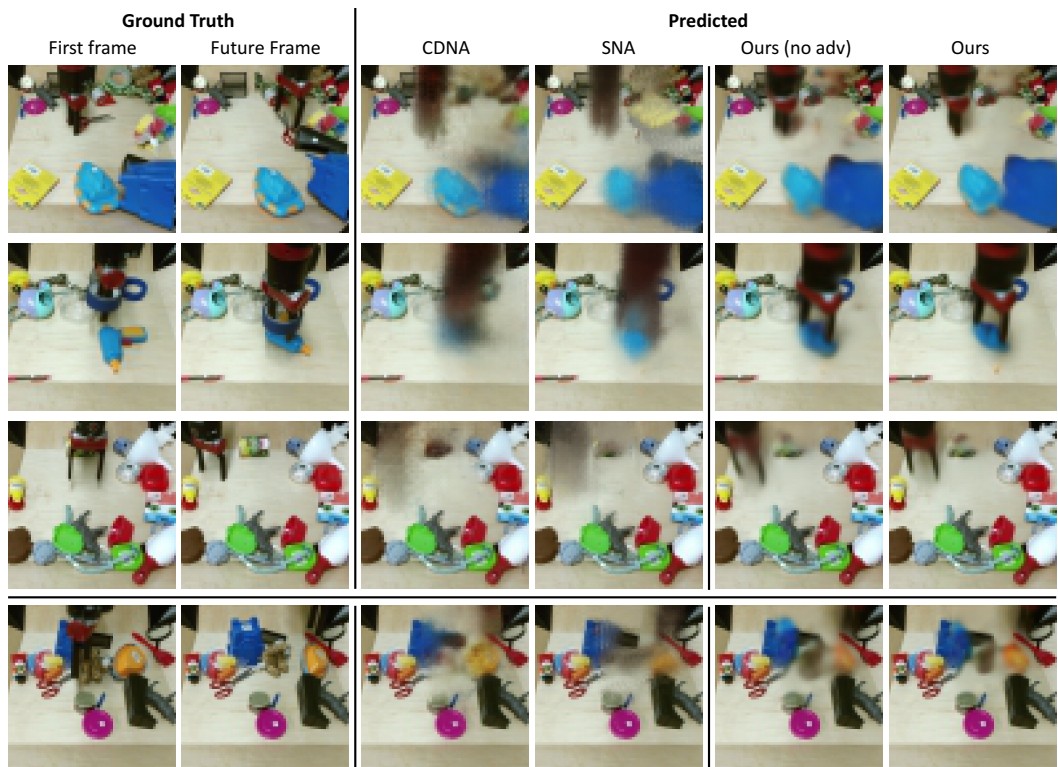

Figure 5: **Qualitative examples of visual quality.** We show selected examples of predicted frames. The left two columns show the initial ground truth frame, as well as a frame in the future. Note that for each method, a prediction is made at each time step. We show the result from a single frame for visual clarity clarity. Please see our supplemental material for a complete set of video results. The examples shown here vary from 6 to 14 steps in the future. The next two columns show results from previous methods **CDNA** (Finn et al., 2016) and **SNA** (Ebert et al., 2017), respectively. These methods produce large blurriness in the gripper and objects the gripper may have come into contact with. The second to last column shows our method, without the adversarial loss. As seen in the first three rows, our improved architecture is able to produce a sharper image, even with an $\ell_2$ regression loss. The final column shows our full method (**Ours**), with the adversarial loss. Note the increased sharpness on the gripper in the first three rows, along with the blue objects in the first and second row, relative to **Ours (no adv)**. The last row shows a failure case, where our method is unable to recover high frequencies and produces a blurry, visually implausible, result.

Qualitative results for the predictions of each model are shown in Figure 5. Further results, including illustrations of motion prediction for a moving point on an object, are shown in Appendix A.

## 5    CONCLUSION

We proposed a method for multi-step video prediction that is focused on both accurate modeling of object motion and the prediction of naturalistic images. We demonstrate that our model can effectively model complex physical interactions between a robotic arm and objects in its environment, without any additional supervision aside from raw image pixels and actions. Our model implicitly predicts object motion by means of a flow-based frame predictor, which computes transformations from the previous frame to the next and then applies them to generate the predicted pixels. This flow can be used to predict the motions of individual points in the environment, and we further propose a new evaluation metric for such models based on the accuracy of the prediction motions as compared to ground truth labels. Furthermore, our model incorporates an adversarial loss, which further improves the realism of the snythesized video predictions. Our evaluation with real human judges indicates that both the improved flow-based architecture and the adversarial loss help to produce more realistic images, and that our method substantially outperforms prior techniques (Finn et al., 2016; Ebert et al., 2017) in terms of prediction quality.

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

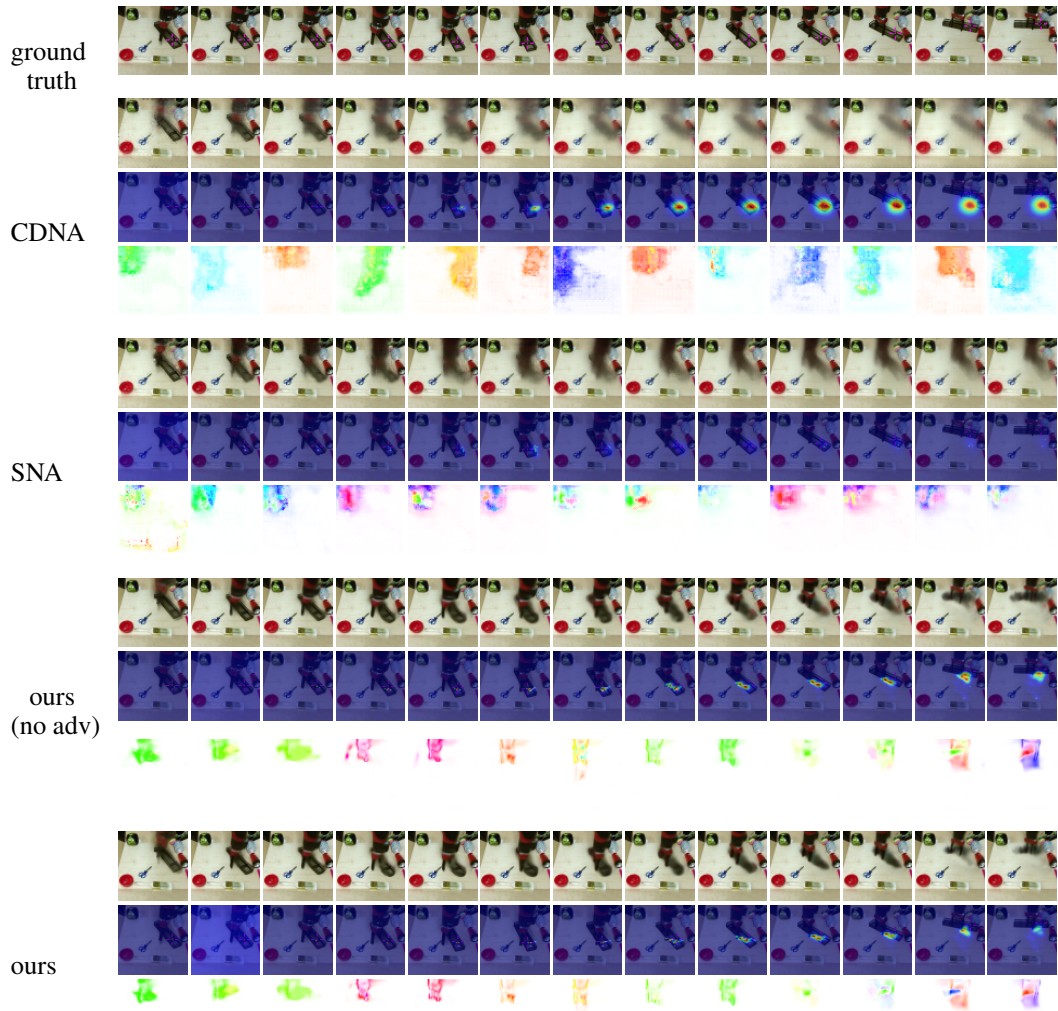

ground truth

CDNA

SNA

ours (no adv)

ours

Figure 6: Predicted frames, pixel distributions, and flows. None of the model are trained using optical flow supervision – only supervision from raw pixels.

## A    FURTHER VISUALIZATIONS OF THE RESULTS

In Figure 6, we show additional visualizations of the video predictions and the implicit object position and flow predictions.

