# OpenReview forum: "Self-Supervised Learning of Object Motion Through Adversarial Video Prediction"
_ICLR.cc/2018/Conference — Reject_

### Official Review · AnonReviewer3 · 2017-11-25
**fine paper, but the authors should aim higher**

**Rating:** 7
**Confidence:** 5

**Review:**

This is a fine paper that generally reads as a new episode in a series on motion-based video prediction with an eye towards robotic manipulation [Finn et al. 2016, Finn and Levine 2017, Ebert et al. 2017]. The work is rather incremental but is competently executed. It is in line with current trends in the research community and is a good fit for ICLR. The paper is well-written, reasonably scholarly, and contains stimulating insights.

I recommend acceptance, despite some reservations. My chief criticism is a matter of research style: instead of this deluge of barely distinguishable least-publishable-unit papers on the same topic, in every single conference, I wish the authors didn’t slice so thinly, devoted more time to each paper, and served up a more substantial dish.

Some more detailed comments:

- The argument for evaluating visual realism never quite gels and is not convincing. The paper advocates two primary metrics: accuracy of the predicted motion and perceptual realism of the synthesized images. The argument for motion accuracy is clear and is clearly stated: it’s the measure that is actually tied to the intended application, which is using action-conditional motion prediction for control. A corresponding argument for perceptual realism is missing. Indeed, a skeptical reviewer may suspect that the authors needed to add perceptual realism to the evaluation because that’s the only thing that justifies the adversarial loss. The adversarial loss is presented as the central conceptual contribution of the paper, but doesn’t actually make a difference in terms of task-relevant metrics. A skeptical perspective on the paper is that the adversarial loss just makes the images look prettier but makes no difference in terms of task performance (control). This is an informative negative result. It's not how the paper is written, though.

- The “no adversary”/“no adv” condition in Table 1 and Figure 4 is misleading. It’s not properly controlled. It is not the case that the adversarial loss was simply removed. The regression loss was also changed from l_1 to l_2. This is not right. The motivation for this control is to evaluate the impact of the adversarial loss, which is presented as the key conceptual contribution of the paper. It should be a proper control. The other loss should remain what it is in the full “Ours” condition (i.e., l_1).

- The last sentence in the caption of Table 1 -- “Slight improvement in motion is observed by training with an adversary as well” -- should be removed. The improvement is in the noise.

- Generally, the quantitative impact of the adversarial loss never comes together. The only statistically significant improvement is on perceptual image realism. The relevance of perceptual image realism to the intended task (control) is not substantiated, as discussed earlier.

- In the perceptual evaluation procedure, the “1 second” restriction is artificial and makes the evaluated methods appear better than they are. If we are serious about evaluating image realism and working towards passing the visual Turing test, we should report results without an artificial time limit. They won’t look as flattering, but will properly report our progress on this journey. If desired, the results of timed comparisons can also be reported, but reporting just a timed comparison with an artificial limit of 1 second may mislead some readers into thinking that we are farther along than we actually are.


There are some broken sentences that mar an otherwise well-written paper:

- End of Section 1, “producing use a learned discriminator and show improvements in visual quality”

- Beginning of Section 3, “We first present the our overall network architecture”

- page 4, “to choose to copy pixels from the previous frame, used transformed versions of the previous frame”

- page 4, “convolving in the input image with”

- page 5, “is know to produce”

- page 5, “an additional indicating”

- page 5, “Adam Kingma & Ba (2015)” (use the other cite command)

- page 5, “we observes”

- page 5, “smaller batch sizes degrades”

- page 5, “larger batch sizes provides”

---

### Official Review · AnonReviewer1 · 2017-11-27
**Incremental contributions and incomplete evaluations**

**Rating:** 3
**Confidence:** 4

**Review:**

This paper is concerned with video prediction, for use in robotic motion planning. The task is performed on tabletop videos of a robotic arm manipulator interacting with various small objects. They use a prior model proposed in Finn et al. 2016, make several incremental architectural improvements, and use an adversarial loss function instead of an L2 loss. They also propose a new metric, motion accuracy, which uses the accuracy of the predicted position of the object instead of conventional metrics like PSNR, which is more relevant for robotic motion planning.

They obtain significant quantitative improvements over the previous 2 papers in this domain (video prediction on tabletop with robotic arm and objects) on both type of metrics - image assessment and motion accuracy. They also evaluate realism images using AMT fooling - asking turks to chose the fake between between real and generated images, and obtain substantial improvements on this metric as well.

A major point of concern is that they do not use the public dataset proposed in Finn et al. 2016, but use their own (smaller) dataset. They do not mention whether they train the previous methods on the new dataset, and some of their reported improvements may be because of this. They also do not report results on unseen objects, when occlusions are present, and on human motion video prediction, unlike the other papers.

The adversarial loss helps significantly only with AMT fooling or realism of images, as expected because GANs produce sharp images rather than distributions, and is not very relevant for robot motion planning. The incremental architectural changes, different dataset and training are responsible for most of the other improvements.

---

### Official Review · AnonReviewer4 · 2017-12-04

**Rating:** 3
**Confidence:** 5

**Review:**


1) Summary
This paper proposes a flow-based neural network architecture and adversarial training for multi-step video prediction. The neural network in charge of predicting the next frame in a video implicitly generates flow that is used to transform the previously observed frame into the next. Additionally, this paper proposes a new quantitative evaluation criteria based on the observed flow in the prediction in comparison to the groundtruth. Experiments are performed on a new robot arm dataset proposed in the paper where they outperform the used baselines.


2) Pros:
+ New quantitative evaluation criteria based on motion accuracy.
+ New dataset for robot arm pushing objects.

3) Cons:
Overall architectural prediction network differences with baseline are unclear:
The differences between the proposed prediction network and [1] seem very minimal. In Figure 3, it is mentioned that the network uses a U-Net with recurrent connections. This seems like a very minimal change in the overall architecture proposed. Additionally, there is a paragraph of “architecture improvements” which also are minimal changes. Based on the title of section 3, it seems that there is a novelty on the “prediction with flow” part of this method. If this is a fact, there is no equation describing how this flow is computed. However, if this “flow” is computed the same way [1]  does it, then the title is misleading.


Adversarial training objective alone is not new as claimed by the authors:
The adversarial objective used in this paper is not new. Works such as [2,3] have used this objective function for single step and multi-step frame prediction training, respectively. If the authors refer to the objective being new in the sense of using it with an action conditioned video prediction network, then this is again an extremely minimal contribution. Essentially, the authors just took the previously used objective function and used it with a different network. If the authors feel otherwise, please comment on why this is the case.


Incomplete experiments:
The authors only show experiments on videos containing objects that have already been seen, but no experiments with objects never seen before. The missing experiment concerns me in the sense that the network could just be memorizing previously seen objects. Additionally, the authors present evaluation based on PSNR and SSIM on the overall predicted video, but not in a per-step paradigm. However, the authors show this per-step evaluation in the Amazon Mechanical Turk, and predicted object position evaluations.


Unclear evaluation:
The way the Amazon Mechanical Turk experiments are performed are unclear and/or not suited for the task at hand.
Based on the explanation of how these experiments are performed, the authors show individual images to mechanical turkers. If we are evaluating the video prediction task for having real or fake looking videos, the turkers need to observe the full video and judge based on that. If we are just showing images, then they are evaluating image synthesis, which do not necessarily contain the desired properties in videos such as temporal coherence.


Additional comments:
The paper needs a considerable amount of polishing.


4) Conclusion:
This paper seems to contain very minimal changes in comparison to the baseline by [1]. The adversarial objective is not novel as mentioned by the authors and has been used in [2,3]. Evaluation is unclear and incomplete.


References:
[1] Chelsea Finn, Ian Goodfellow, and Sergey Levine. Unsupervised learning for physical interaction through video prediction. In NIPS, 2016.
[2] M. Mathieu, C. Couprie, and Y. LeCun. Deep multi-scale video prediction beyond mean square error. In ICLR, 2016.
[3] Ruben Villegas, Jimei Yang, Seunghoon Hong, Xunyu Lin, Honglak Lee. Decomposing Motion and Content for Natural Video Sequence Prediction. In ICLR, 2017

---

### Official Review · AnonReviewer2 · 2017-12-05
**Marginal contributions and missing comparison with state of the art**

**Rating:** 3
**Confidence:** 5

**Review:**

In this paper a neural-network based method for multi-frame video prediction is proposed. It builds on the previous work of [Finn et al. 2016] that uses a neural network to predict transformation parameters of an affine image transformation for future frame prediction, an idea akin to the Spatial Transformer Network paper of [Jaderberg et al., 2015]. What is new compared to [Finn et al. 2016] is that the authors managed to train the network in combination with an adversarial loss, which allows for the generation of more realistic images. Time series modelling is performed via convolutional LSTMs. The authors evaluate their method based on a mechanical turk survey, where humans are asked to judge the realism of the generated images; additionally, they propose to measure prediction quality by the distance between the manually annotated positions of objects within ground truth and predicted frames.

My main concerns with this paper are novelty, reproducibility and evaluation.

* Novelty. The network design builds heavily on the work of [Finn et al., 2106]. A number of design decisions (such as instance normalization) seem to help yield better results, but are minor contributions. A major contribution is certainly the combination with an adversarial loss, which is a non-trivial task. However, the authors claim that their method is the first to combine multi-frame video prediction with an adversarial loss, which is not true. A recent work, presented at CVPR this year also does multi-frame prediction featuring an adversarial loss and explicitly models and captures the full dense optical flow (though in the latent space) that allows non-trivial motion extrapolation to future frames. This work is neither mentioned in the related work nor compared to.

Lu et al. , Flexible Spatio-Temporal Networks for Video Prediction, CVPR 2017

This recent work builds on another highly relevant work, that is also not mentioned in the paper:

Patraucean et al. Spatio-temporal video autoencoder with differentiable memory, arxiv 2017

Since this is prior state-of-the-art and directly applicable to the problem, a comparison is a must.

* Reproducibility and evaluation
The description of the network is quite superficial. Even if the authors released their code used for training (which is not mentioned), I think the authors should aim for a more self-contained exposition. I doubt that a PhD student would be able to reimplement the method and achieve comparable results given the paper at hand only. It is also not mentioned whether the other methods that the authors compare to are re-trained on their newly proposed training dataset. Hence, it remains unclear to what extend the achieved improvements are due to the proposed network design changes or the particular dataset they use for training. The authors also don't show any results on previous datasets, which would allow for a more objective comparison to existing state of the art. Another point of criticism is the way the Amazon Mechanical Turk evaluation was performed. Since only individual images were shown, the evaluation mainly measures the quality of the generated images. Since the authors combine their method with a GAN, it is not surprising that the generated images look more realistic. However, since the task is *video* prediction, it seems more natural to show small video snippets rather than individual images, which would also evaluate temporal consistency.

* Further comments:
The paper contains a number of broken sentences, typos and requires a considerable amount of polishing prior to publication.

---

### Public Comment · ~Oleksandr_Zaytsev1 · 2017-11-09
**Requesting access to the dataset**

Hello,
we want to take part in the ICLR 2018 Reproducibility Challenge (http://www.cs.mcgill.ca/~jpineau/ICLR2018-ReproducibilityChallenge.html) and replicate the experiments described in this paper. Would it be possible to access the trajectories dataset that was described in a paper and used for training?

Thank you!

---

> ### Public Comment · (anonymous) · 2017-11-12
> **Links to publicly available robot pushing datasets**
>
> The dataset will be released upon publication. In the meantime, there are similar datasets that are publicly available such as the following:
> https://sites.google.com/site/brainrobotdata/home/push-dataset
> https://sites.google.com/view/sna-visual-mpc

---

### Decision · Program_Chairs · 2018-01-29
**ICLR 2018 Conference Acceptance Decision**

**Decision:**

Reject

**Comment:**

The paper proposes adversarial flow-based neural network architecture with adversarial training for video prediction. Although the reported experimental results are promising, the paper seems below ICLR threshold due to limited novelty and issues in evaluation (e.g., mechanical turk experiment). No rebuttal was submitted.